# Tensorweave 1.0: Interpolating geophysical tensor fields with spatial neural networks

Akshay V. Kamath<sup>1</sup>, Samuel T. Thiele<sup>1</sup>, Hernan Ugalde<sup>2</sup>, Bill Morris<sup>3</sup>, Raimon Tolosana-Delgado<sup>1</sup>, Moritz Kirsch<sup>1</sup>, and Richard Gloaguen<sup>1</sup>

**Correspondence:** Akshay V. Kamath (a.kamath@hzdr.de)

Abstract. Tensor fields, as spatial derivatives of scalar or vector potentials, offer powerful insight into subsurface structures in geophysics. However, accurately interpolating these measurements—such as those from full-tensor potential field gradiometry—remains difficult, especially when data are sparse or irregularly sampled. We present a physics-informed spatial neural network that treats tensors according to their nature as derivatives of an underlying scalar field, enabling consistent, high-fidelity interpolation across the entire domain. By leveraging the differentiable nature of neural networks, our method not only honours the physical constraints inherent to potential fields but also reconstructs the scalar and vector fields that generate the observed tensors. We demonstrate the approach on synthetic gravity gradiometry data and real full-tensor magnetic data from Geyer, Germany. Results show significant improvements in interpolation accuracy, structural continuity, and uncertainty quantification compared to conventional methods.

<sup>&</sup>lt;sup>1</sup>Helmholtz-Zentrum Dresden-Rossendorf, Helmholtz Institute Freiberg, Chemnitzer Str. 40, 09599 Freiberg, Germany.

<sup>&</sup>lt;sup>2</sup>DIP Geosciences, Hamilton, ON Canada.

<sup>&</sup>lt;sup>3</sup>Morris Magnetics Inc., Fonthill, ON Canada.

## 10 1 Introduction

Full-tensor gravity and magnetic gradiometry measurements capture the spatial derivatives of potential fields, offering rich detail about subsurface density and magnetisation variations. These tensor fields enhance geological imaging by encoding directional and gradient information that scalar fields do not straightforwardly provide (Brewster, 2011; Ugalde et al., 2024). However, gradiometry data are typically sparse and anisotropically sampled—often along sub-parallel flight lines—posing significant challenges for downstream analysis, which relies on gridded representations.

Interpolating these tensor fields accurately is far from trivial. Conventional methods often treat tensor components as independent scalar fields, leading to noise amplification, loss of directional trends, and violations of physical constraints like symmetry and harmonicity. More advanced approaches, such as eigen-decomposition-based interpolation (Fitzgerald et al., 2012; Satheesh et al., 2023), attempt to preserve tensor structure, but remain limited in generalisability and scalability.

Recent advances in neural fields (Xie et al., 2022)—also known as coordinate-based or implicit neural representations—offer a promising alternative. These models learn continuous functions that map spatial coordinates to field values. Their differentiable nature allows them to incorporate gradient information directly into training—a key advantage for geophysical applications where tensor data often represents derivatives of an underlying field (Raissi et al., 2019). However, standard multilayer perceptron (MLP) architectures suffer from spectral bias (Rahaman et al., 2019), meaning they struggle to capture high-frequency features common in geophysical signals. To address this, techniques like Random Fourier Feature mapping (Tancik et al., 2020), periodic activation functions (e.g. SiREN, Sitzmann et al., 2020), and wavelet activations (Saragadam et al., 2023) have been introduced, enabling neural fields to model fine-scale spatial variations more effectively.

In this paper, we introduce a physics-informed neural field approach tailored for interpolating geophysical tensor data. Our model learns a single scalar potential field from sparse tensor measurements, leveraging RFF mappings and embedded physical constraints (e.g., symmetry, Laplacian properties) to reconstruct consistent, physically meaningful tensor fields. We further introduce an ensemble strategy to quantify uncertainty in the interpolations, offering insights into data sensitivity and model confidence. We demonstrate this framework on both synthetic gravity gradiometry data and real airborne magnetic gradiometry from Geyer, Germany, highlighting clear improvements over traditional methods in accuracy and structural continuity, as well as opening the door to uncertainty quantification.

## 35 2 Background

A tensor is an algebraic object that encodes multilinear relationships between sets of vectors and linear functionals (Lee, 2012). A tensor field assigns a tensor to every point in space, describing the local structure of a vector field or scalar potential throughout a region. In geophysical applications, tensors naturally arise as derivatives of vector and scalar fields, extending classical multivariable calculus into field-based formulations.

## 40 2.1 Potential fields

Many measured geophysical quantities, such as gravitational acceleration g and the magnetic field b, are conservative vector fields—i.e., they are gradients of scalar potential fields (Blakely, 1995). Within  $\mathbb{R}^3$ , a conservative vector field v is irrotational at all points (given by the position vector v), satisfying

$$\mathbf{v} = \nabla \phi \quad \leftrightarrow \quad \nabla \times \mathbf{v} = 0$$
 (1)

Where  $\nabla = \left[\frac{\partial}{\partial r_x}, \frac{\partial}{\partial r_y}, \frac{\partial}{\partial r_z}\right]$  is the gradient operator. For instance, the magnetic field can be expressed as the gradient of a scalar magnetic potential in regions free of electric currents—a condition typically met outside source distributions. Taking the gradient of v yields the Hessian tensor  $\mathbf{H}$ , a second-order tensor that captures the local curvature of the scalar potential

$$\mathbf{H} = \nabla \mathbf{v} = \nabla (\nabla \phi) = \frac{\partial^2 \phi}{\partial r_i \partial r_j} \equiv \partial_i \partial_j \phi \quad \forall i, j = 1, 2, 3$$
 (2)

In source-free regions, these fields are not only irrotational but also solenoidal—i.e., divergence-free. The divergence of v corresponds to the trace of the Hessian, which reflects the Laplace equation

$$\nabla^2 \phi \equiv \operatorname{tr}(\mathbf{H}) = 0 \tag{3}$$

This implies that, outside source regions, scalar potentials are harmonic functions, and their Hessians are traceless. Additionally, since mixed partial derivatives commute (by Schwarz's theorem), the Hessians are symmetric and thus comprise five independent components.

## 55 2.2 Full Tensor Gradiometry

Direct measurements of second-order Hessian tensors—particularly gravity and magnetic gradient tensors—represent the current frontier in potential field surveying (Rudd et al., 2022; Stolz et al., 2021). Access to the full tensor enables characterisation of scalar field curvature, aiding in tasks such as edge detection, structure delineation (Zuo and Hu, 2015), and magnetic remanence characterisation (Ugalde et al., 2024). These measurements are typically acquired via airborne surveys, which are highly anisotropic in their sampling: dense along flight lines and sparse across them. Vector fields are frequently reconstructed from tensor components using Fourier-domain transfer functions, which integrate the measured gradients into vector components while suppressing noise (Vassiliou, 1986). Since most downstream analyses, including Fourier-based reconstructions, require gridded tensor and vector fields, interpolation is a critical preprocessing step.

Rudd et al. (2022) note that, in practice, tensor components are often treated as separate scalar fields and interpolated using standard methods like minimum curvature or radial basis functions (RBFs). Several alternative approaches have been proposed to enforce physical or geometric constraints during interpolation. For example, Brewster (2011) uses iterative Fourier-domain transformations, while Fitzgerald et al. (2012) suggest interpolating eigenvalues and eigenvectors separately, using a quaternion-based interpolation technique. In essence, the quaternion interpolation algorithm decomposes the process into two parts: interpolating the eigenvalues and the corresponding eigenvectors. Two of the three eigenvalues are interpolated using

standard schemes (e.g., RBF or minimum curvature), with the third computed such that their sum is zero—a direct consequence of the Hessian's traceless-ness. The eigenvector matrix of any symmetric real matrix is guaranteed to be real and orthogonal, allowing it to be represented as a 3D orientation and encoded as a quaternion (Hamilton, 1844), provided some constraints on ordering and sign convention are imposed (Satheesh et al., 2023). These quaternions are then interpolated using Spherical Linear Interpolation or SLERP (Shoemake, 1985), which ensures smooth variation of orientation across space. While SLERP works for two quaternions, Markley et al. (2007) devised a scheme that works across a set of weighted quaternions.

Another widely used approach for interpolating and transforming potential-field (and gradient) data is the equivalent-source / equivalent-layer method: one replaces the true 3D distribution of sources by a 2D layer of hypothetical monopoles or dipoles beneath the observation surface whose field exactly reproduces the measured data on that surface (Dampney, 1969; Blakely, 1995). In practice the infinite layer is discretised into a finite set of sources and the corresponding dense sensitivity matrix is solved—often with regularisation—to obtain source strengths that honour the physical constraints of potential fields and enable stable continuation and derivative transforms (Hansen and Miyazaki, 1984; Blakely, 1995; Oliveira Junior et al., 2023). This formulation is powerful but computationally demanding for large surveys. Consequently, a substantial literature focuses on accelerating the method by exploiting data geometry and matrix structure: scattered equivalent-source gridding (Cordell, 1992); the "equivalent data" concept to reduce system size (Mendonça and Silva, 1994); wavelet compression and adaptive meshing to sparsify sensitivities (Li and Oldenburg, 2010; Davis and Li, 2011); fast iterative schemes in the space/wavenumber domains (Xia and Sprowl, 1991; Siqueira et al., 2017); and scalable algorithms that leverage the block-Toeplitz Toeplitz-block (BTTB) structure of the sensitivity matrix (Piauilino et al., 2025). Recent machine-learning-inspired variants (e.g., gradient-boosted equivalent sources) further cut memory and runtime for continental-scale datasets (Soler and Uieda, 2021). Open-source implementations, notably *Harmonica*, provide practical tools for gravity and magnetic datasets using these ideas (Fatiando a Terra Project et al., 2024).

However, these methods still have limitations: component-wise methods can be insensitive to the true shape of the tensor, whereas full-tensor schemes involve complex handling of 3D rotations, which are complicated due to the existence of indeterminate points and the need for shifting reference frames due to non-uniqueness of 3D rotations. Equivalent source methods offer a physically consistent approach, but suffer from the trade-off between computational expense and fidelity of the interpolated result (e.g., Piauilino et al., 2025). In this contribution, we propose a neural field method that interpolates the scalar potential field directly—constrained by physical laws and Hessian measurements—to produce consistent, noise-minimising tensor and vector fields that respect observations and preserve geologically meaningful structures.

## 2.3 Neural fields

Neural fields—also known as implicit neural representations, or spatial neural networks—are models that represent continuous spatial functions using neural networks. Unlike traditional methods that store information in discrete grids or meshes, neural fields encode spatial structure within the weights and biases of a neural network, enabling resolution-independent, continuous representations of complex signals.

The application of spatial neural networks in geoscience dates back to Openshaw (1993), who used them for interpolating sparse spatial data and found their performance competitive with fuzzy logic and genetic algorithms, a conclusion also reached by Hewitson et al. (1994). More recently, neural fields have gained traction in computer vision—for example, in volumetric radiance field modelling (Mildenhall et al., 2020)—and in geoscience applications such as 3D geological modelling (Hillier et al., 2023) and potential field representation (Smith et al., 2025).

A key advantage of neural fields is their differentiability: they allow access not only to predicted signals but also to their spatial derivatives via automatic differentiation. This is especially useful when the scalar field itself is unmeasurable or physically arbitrary, but its gradients are measurable—as is often the case in geological modelling using structural orientation data (Kamath et al., 2025; Thiele et al., 2025), or in reconstructing potential fields from tensor gradiometry data.

## 3 Methodology

105

115

120

This section outlines the key components of our proposed framework, including the use of random Fourier features, a harmonic feature embedding, model architecture, training regimen, and loss function. We also describe the methodology used to generate the synthetic dataset used in our study.

## 3.1 Random Fourier Feature mapping

A common challenge in implicit neural representations is the mismatch between low-dimensional input coordinates and the complex, high-frequency structure of the target signal. To address this, we employ *Random Fourier Feature* (RFF) mapping—a kernel approximation technique introduced by Rahimi and Recht (2007) and adapted to deep learning by Tancik et al. (2020). RFF mapping projects spatial coordinates into a higher-dimensional space, making it easier for the network to learn fine-scale spatial variation.

Given the position vector  $r \in \mathbb{R}^N$ , we define a frequency (also called weights) matrix  $\mathbf{W}$ , of the dimension  $M \times N$ , with every component sampled from a standard Gaussian distribution, where M is the number of Fourier features (i.e., frequencies). To encode known signal frequency characteristics (e.g., the maximum possible frequency based on sampling resolution), we rescale the weights matrix using different length scales. Therefore, for a 3D input, we get a 2M dimensional feature vector  $\boldsymbol{\nu}_s$  for every length scale  $\ell_s$  given by

$$\nu_s = [\sin(2\pi \mathbf{W}_s \mathbf{r}), \cos(2\pi \mathbf{W}_s \mathbf{r})], \text{ where } \mathbf{W}_s = \ell_s^{-1} \mathbf{W}^{M \times 3}$$
 (4)

Where  $\mathbf{r} = [x, y, z]^T$ ,  $W_{ij} \sim \mathcal{N}(0, 1)$ ,  $\sin(\mathbf{x}) := [\sin(x_i)]_i$ , and [:,:] represents concatenation along the feature axis. Hence, for L length scales, as the feature vectors are concatenated, we get a 2ML-dimensional feature vector. This feature vector is then fed into the subsequent multi-layer perceptron to get the scalar potential at the input coordinate.

The transformation enables the model to capture high-frequency details more effectively, while the random sampling of frequencies introduces a useful stochastic component. When followed by a linear MLP with no non-linear activations, the

resulting mapping approximates a full Fourier reconstruction of the signal (Bracewell and Kahn, 1966). Non-linear activations help the model fit sparse data more flexibly (LeCun et al., 2015), albeit at the cost of simplicity, interpretability, and gradient stability.

## 3.2 Harmonic feature embedding

Applying Fourier features uniformly in all dimensions can hinder convergence when modelling harmonic fields. By Liouville's Theorem (Axler et al., 2001), any bounded harmonic function on  $\mathbb{R}^N$  is constant, so naive periodic embeddings can bias the network toward trivial solutions. We therefore introduce a *harmonic* Fourier mapping that uses Fourier features in the horizontal plane and an analytically motivated vertical decay. Since our model reconstructs the spatial domain signal from a combination of sinusoids, we use the frequencies of the sinusoids to encode the harmonicity condition into the mapping. Specifically, in the Fourier domain, if we couple Laplace's equation with the standard separation-of-variables method for a scalar potential  $\phi$ , we get an ordinary differential equation of the form

$$(k_z^2 - \|\mathbf{k}_h\|_2^2) \mathcal{F}_{x,y}(\phi) = 0 \tag{5}$$

Where  $\mathcal{F}_{x,y}(\cdot)$  is the 2D Fourier transform operator,  $\mathbf{k}_h = [k_x, k_y]$  represents the horizontal wavenumbers (i.e., frequencies), and  $k_z$  is the vertical wavenumber (Lacava, 2022). This implies that the vertical wavenumber is constrained by  $k_z^2 = k_x^2 + k_y^2$ , making the vertical dependence evanescent and not oscillatory. This is the classical half-space solution of Laplace's equation and underpins upward/downward continuation in potential-field geophysics (Blakely, 1995; Parker, 1973). Our harmonic embedding scheme simply bakes in the same physics at a feature-level, helping with convergence while training on mostly co-planar datasets, and potentially allowing robust upward/downward continuation.

For a 3D input  $\mathbf{r} = [x, y, z]^T$ , we extract the horizontal coordinates  $\mathbf{r}_{xy} = [x, y]^T$ . As defined in the previous section, we generate a random weights matrix  $\mathbf{W}^{M\times 2}$  with the entries independently drawn from a standard normal distribution  $W_{ij} \sim \mathcal{N}(0, 1)$ , and scale it with the length scale  $\ell_s$  to acquire a scaled matrix  $\mathbf{W}_s = \ell_s^{-1} \mathbf{W}$ . For every length scale, we define a new vector,  $\kappa_s$ , such that

155 
$$(\kappa_s)_m = \|\mathbf{W}_{s,m:}\|_2, \quad \kappa_s \in \mathbb{R}^M$$

Where m refers to the mth row of the  $W_s$  matrix, giving us a vector of length M i.e., one norm per row (per feature). With element-wise sine/cosine and Hadamard product  $\odot$ , the new feature vector  $\boldsymbol{\nu}_s$  for the scale  $\ell_s$  is

$$\nu_s = \left[ \sin(2\pi \mathbf{W}_s r_{xy}) \odot e^{-\kappa_s z}, \cos(2\pi \mathbf{W}_s r_{xy}) \odot e^{-\kappa_s z} \right]$$
(7)

where z is the vertical coordinate, and the exponential is applied element-wise, producing a 2M-dimensional vector. Concatenating across L scales yields a 2ML-dimensional embedding that encodes horizontal oscillations with physically consistent vertical decay, aligning the features with solutions of Laplace's equation and improving generalisation in undersampled regions.

## 3.3 Synthetic dataset

To evaluate our method, we generated a synthetic gravity gradiometry dataset (Fig. 1) using SimPEG (Cockett et al., 2015).

The model consists of three density-contrast anomalies within a zero-density half-space:

- 1. Torus: +1 g cm<sup>-3</sup>, semi-major axis 450 m, semi-minor axis 220 m, cross-section radius 40 m, lying in the xy plane and rotated  $12^{\circ}$  CCW from the y-axis.
- 2. Dyke:  $+0.15 \text{ g cm}^{-3}$ , 60 m aperture, striking at  $45^{\circ}$  to the y-axis.
- 3. Cube:  $-0.2 \text{ g cm}^{-3}$ , 400 m side length, rotated  $45^{\circ}$  about the vertical (z) axis.
- The simulation mesh has a voxel size of 20 m. This geometry offers a challenging mix of sharp discontinuities and smooth curvature for testing interpolation. We generated a high-resolution ground truth dataset sampled at 25 m spacing both along and across the lines, as well as a low-resolution airborne-style dataset with flight lines 200 m apart in the y direction (perpendicular to the flight line), and sampled densely (15 m) along the x direction (Fig. 1b-f). Furthermore, to make the data more realistic, We corrupt the full-tensor gradiometry data with additive white Gaussian noise at a prescribed signal-to-noise ratio (SNR). For each independent component of the Hessian  $H_k$  (where  $k \in \{xx, xy, xz, yy, yz\}$ ), we estimate the power spectrum  $P_k = \langle H_k^2 \rangle$ , convert the target SNR from dB to linear units as:

$$SNR_k = 10^{SNR_{k,dB}/10}$$
(8)

Using this SNR, we set the noise variance of the signal as  $\sigma_k^2 = \frac{P_k^2}{\text{SNR}_k}$ . Independent samples  $\varepsilon_{i,k} \sim \mathcal{N}(0, \sigma_k^2)$  are then added to each datum, to acquire a noisy synthetic dataset  $(\tilde{H})$ , given by:

80 
$$\tilde{H}_{i,k} = H_{i,k} + \varepsilon_{i,k} \quad \forall i = 1...P$$
 (9)

Where P is the number of points in the dataset.

To test robustness to data sparsity, we also computed 10 versions of the low-resolution dataset with line spacings varying from  $80\,\mathrm{m}$  to  $560\,\mathrm{m}$ . These were used to benchmark interpolation quality and information loss under varying acquisition densities. Comparisons were made with a truncated RBF interpolator (250 nearest neighbours, smoothing factor 100), as well as results from the quaternion interpolation (QUAT; Fitzgerald et al., 2012), combining RBF-interpolated eigenvalues with SLERP-interpolated quaternions. All results were evaluated on the same high-resolution grid using the  $R^2$  (coefficient of determination), MSE (Mean Squared Error), and SSIM (Structural Similarity Index Measure).

## 3.4 Model architecture, training regimen, and loss functions

Our model has two main components: a RFF mapping block followed by a sequence of fully-connected feed-forward layers that together produce a continuous scalar field representation (Fig. 2). In our tests, we varied the number of Fourier features from anywhere between 16 to 64, depending on the complexity of underlying field. The specifications of each individual model showcased in this contribution can be found in the following sections.

The MLP block in our model uses non-linear activations for all layers except the output layer. As our framework involves computing second derivatives with AD, activation functions like ReLU (which do not satisfy the  $C^2$  differentiability criterion) resulted in abrupt edges within the resultant interpolation. Notably, even within the activations that satisfy the aforementioned criterion, some functions performed better than the others. For example, the Hyperbolic Tangent activation function has extremely small second order derivatives which tend to get saturated, impeding convergence. These activations are stable, but not ideal for our models. Among the various activation functions tested, Swish (Sigmoid Linear Unit, SiLU; Ramachandran et al., 2017) and Mish (Misra, 2019) activations provided the best results.

195

The loss function used to train our model involves two types of losses - a data loss and a Laplacian loss. The data loss is computed at the points of measurement between the measured tensor components and the Hessians acquired from the predicted scalar field through automatic differentiation (AD) (Margossian, 2019). Since the model is built with *PyTorch* (v2.8.0; Paszke et al., 2019), we use the inbuilt *autograd* engine to compute Hessians from the scalar field output. For the predicted scalar field  $\phi$ , the data loss term is given by

$$\mathcal{L}_d = \frac{1}{P} \sum_{p=1}^{P} \left| \partial_i \partial_j \phi^p - H_{ij}^p \right| \quad \forall i, j = 1, 2, 3$$

$$\tag{10}$$

Where  $\partial_i \phi^p$  refers to the partial derivative with the respect to the *i*th input computed with AD at the *p*th location, and  $\mathbf{H}^p$  is the corresponding measured hessian tensor. The first, second, and third indices correspond to x (East-West), y (North-South) and z (Up-Down) axes respectively. Only the upper-triangular part of the Hessian is used for loss computation i.e., the losses for the off-diagonal components are only considered once per measurement. Hence we get a six-component data loss vector, consisting of the misfit between the xx, xy, yz, yy, yz and zz components.

The second term in the loss function is derived to encourage the predicted scalar field to conform with a partial differential equation defined across the whole domain of interest. Since the predicted field has to be harmonic not only at the points of measurement, but everywhere, we thus penalise non-zero traces of the predicted Hessian tensors, hereafter referred to as the Laplacian loss. During every training epoch, we use a Poisson disk sampling routine based on a hierarchical dart throwing approach (White et al., 2007), to select evenly spaced points within a predefined grid that covers the area of interest. The spacing for these points is evaluated using an exponentially decaying function that goes from a user specified large spacing (at the start of training) to a tighter spacing (towards the final epochs). For the Q sampled points in any epoch, the Laplacian loss is given by:

$$\mathcal{L}_{l} = \frac{1}{Q} \sum_{q=1}^{Q} |\operatorname{tr}(\nabla(\nabla \phi^{q}))| = \frac{1}{Q} \sum_{q=1}^{Q} |\partial_{i} \partial_{i} \phi^{q}|$$
(11)

Here, the Einstein summation convention is used to represent the Laplacian, and the superscript refers to the point of evaluation. This loss penalises high values of the trace of the predicted hessian tensor outside the measured points, thereby encouraging harmonicity on the underlying scalar field within the domain of interest. Hence, for every epoch, we get a seven component combined loss vector, with the first six components corresponding to the data loss, and the seventh component referring to the global Laplacian loss.

When combined, the total loss acquired from equations 10 and 11 is thus

$$\mathcal{L}_{total} = \sum_{i=1}^{7} \alpha_i \mathcal{L}_i \tag{12}$$

Where  $\mathcal{L}_i$  is the *i*th component of the combined loss vector,  $[\alpha_i]$  are the corresponding hyperparameters. Instead of manually fine-tuning these hyperparameters, we tested various multi-objective optimisation schemes for our framework. The best-performing scheme involved dynamically updating hyperparameters, such that every loss function was scaled by the real-time magnitude of the loss. Mathematically, we replace the  $[\alpha_i]$  in Eq. 12 as follows:

$$\mathcal{L}_{total} = \sum_{i=1}^{7} \frac{\mathcal{L}_i}{\tilde{\mathcal{L}}_i} \tag{13}$$

Where  $\tilde{\mathcal{L}}_i$  refers to the magnitude of the loss, detached from the computational graph. This ensures that the gradients only flow through the numerator of the scaled loss, even when the real-time value of the loss is always equal to 1. This real-time normalisation yields a scale-invariant objective whose update is approximately  $\nabla_{\theta} \sum_{i} \log \mathcal{L}_i$  (where  $\nabla_{\theta}$  is the gradient with respect to the network parameters), encouraging proportional improvements across terms. Similar in spirit to uncertainty-based weighting (Kendall et al., 2017), *GradNorm* (Chen et al., 2017), and Density Weight Averaging (Liu et al., 2019), this variant requires no extra parameters or gradient-norm computations and worked reliably in our setting.

We train the MLP parameters (weights and biases) with Adam (Kingma and Ba, 2014), while keeping the RFF encoder fixed after initialisation. The initial learning rate is set to  $10^{-3}$  (occasionally  $10^{-2}$  when the initial loss scale is large). We apply a plateau scheduler that reduces the learning rate by a factor of 0.8 whenever the loss fails to improve for 20 epochs. Optionally, we also optimise a set of learnable length-scale parameters that modulate the Fourier features; the log-values of these scales are stored as parameters and updated jointly with the MLP. However, the learnable nature of these length scales did not help the model convergence greatly, reproducing results explored by Tancik et al. (2020), which suggested that neural fields fail to suitably optimise these length scale parameters.

To make the stopping criterion robust to small oscillations, we monitor an exponential moving average (EMA) of the pre-scaled loss:

$$\hat{\mathcal{L}}_n = \beta \hat{\mathcal{L}}_{n-1} + (1-\beta)\mathcal{L}_n \quad \beta \in (0,1)$$
(14)

Where  $\hat{\mathcal{L}}_n$  denotes the smoothed loss on the *n*th epoch. This combination—*Adam* for fast, well-scaled updates, plateau-based learning-rate decay, and EMA-stabilised early stopping—follows common best practices for training smooth function approximators and has been shown to curb overfitting while maintaining convergence speed (Goodfellow et al., 2016; Prechelt, 1998; Bottou, 2012).

## 3.5 Uncertainty estimation

A key benefit of using RFF embeddings is that their stochastic nature allows for ensemble-based uncertainty estimation. As a result of the stochasticity, each initialisation of the RFF mapping induces a unique basis in the feature space, causing the

255 neural network to converge on a solution that represents a random sample from a broader distribution of plausible scalar fields conditioned on the training data.

To exploit this property for uncertainty quantification, we generate an ensemble of model outputs by varying the random seed used to sample the RFF projection matrix. Ensemble-based uncertainty quantification has a long and successful history in geophysics, particularly in subsurface modelling and inversion. In seismic full waveform inversion (FWI), ensembles have been used to assess the variability and reliability of recovered velocity models under data noise and model ambiguities (Fichtner et al., 2011). In reservoir geophysics, the Ensemble Kalman Filter (EnKF) has become a widely used tool to propagate uncertainty in dynamic reservoir simulation and history matching (Evensen, 2009). More recently, ensemble-based methods have also been applied to probabilistic gravity and magnetotelluric inversion (Tveit et al., 2020; Giraud et al., 2023), demonstrating their utility in quantifying non-uniqueness and guiding data acquisition strategies.

In our implementation, each ensemble member corresponds to a different realisation of the frequency space, leading to stochastically independent function approximations that depend, largely, on the degree to which the solution is constrained by the available data. This ensemble-based approach provides a Monte Carlo-style estimate of the model's epistemic uncertainty. Furthermore, because the scalar field is modelled continuously, we can propagate this ensemble approach to the field's derivatives, helping us quantify uncertainty in derived physical quantities. Therefore, we showcase our results as the Ensemble Neural Field (ENF) method, which corresponds to the average prediction from an ensemble of models. We also compute results from the individual models within the ensemble (shown as the Neural Field or NF result), to ascertain the effect of averaging over multiple predictions.

## 4 Results

## 4.1 Synthetic Data

We first evaluate the Ensemble Neural Field (ENF) method on the synthetic gravity gradiometry dataset, comparing it against a Truncated Radial Basis Function (RBF) interpolator (Fig. 3). The ensemble showcased here has 25 models, each with 16 Fourier features, three length scales of [200, 400, 1000], and two hidden layers with 256 neurons each. Each model within the ensemble was trained for 400 epochs. A predefined grid with a cell-size of 25 m was provided for evaluating the Laplacian loss, with the Poisson sampling radius going from 250 m to 80 m. Panels (a) and (b) show the residuals between predicted and true *H*<sub>xy</sub> values for the RBF and ENF methods, respectively. The RBF output exhibits high-amplitude residuals (MSE = 4.60 eotvos) between flight lines, indicating overfitting to sampled regions and poor generalisation across them. It also fails to preserve continuity in linear trends that lie at high angles to the flight direction. In contrast, the ENF method yields spatially smoother residuals with significantly lower error (MSE = 0.30 eotvos; improvement of ≈ 93.4% over the RBF), suggesting homogeneous improved performance across the domain. Insets in both panels show 1:1 kernel density plots, where the ENF predictions cluster more tightly along the identity line—further confirming its accuracy.

For a quantitative measure of the improvement offered by our method, we plot the  $R^2$  scores for each tensor component across three interpolation methods: RBF, and two neural field-based (NF and ENF) (Fig. 3c). The NF method reflects the mean  $R^2$  from 25 independently trained models, with error bars showing standard deviation. The ENF method, by contrast, uses the averaged prediction across those same models. Both neural field approaches outperform classical methods, with ENF showing a slight edge—demonstrating that ensemble averaging reduces variance and enhances prediction stability. Fig. 3d shows the loss curves for the various losses for one of the models within the ensemble, as a function of epochs. The data loss terms reasonably plateau after reaching values of  $\approx 1$  eotvos, while the real-time updating hyperparameters help avoid overfitting to a single component. The Laplacian loss (dotted pink line;  $\mathcal{L}_l$ ) keeps steadily decreasing as the sampling gets tighter and ever more points are sampled from the grid.

To further evaluate structural accuracy, we compute the Structural Similarity Index Measure (SSIM) between predicted and true tensor fields (Fig. 4). The ENF method achieves higher SSIM scores across all three components—0.95  $(H_{xx})$ , 0.97  $(H_{xy})$ , and 0.96  $(H_{xz})$ —compared to 0.78, 0.64, and 0.76 for RBF. The greatest improvement is seen in  $H_{xy}$  (improvement of  $\approx 50.46\%$ ), where RBF results show structural distortion, over-smoothing, and "boudinage" artefacts along flight lines (Naprstek and Smith, 2019). ENF, on the other hand, preserves coherent anomalies and directional continuity even across sparsely sampled regions.

## 4.2 Rate of information loss

To assess robustness under sparse sampling, we compare the interpolation results for varying line spacings from  $80\,\mathrm{m}$  to  $560\,\mathrm{m}$  (Fig. 5). Classical methods (RBF and quaternion-based interpolation, or QUAT) show sharp drops in accuracy beyond  $200\,\mathrm{m}$  spacing. For example, the RBF method's root-mean-squared  $R^2$  (computed over the components) drops to 0.54, and the root-mean-squared SSIM plummets to 0.26 at  $560\,\mathrm{m}$ . In contrast, NF interpolation maintains relatively stable performance up to  $\approx 400\,\mathrm{m}$  spacing, with a much gentler decline at wider gaps. At  $560\,\mathrm{m}$ , the NF model still achieves a root-mean-squared  $R^2$  of 0.91 and an SSIM of 0.65.

The MSE trends mirror this behaviour: classical methods exhibit steep error increases with sparser lines, while the NF model degrades more gracefully. QUAT offers minor improvements over component-wise interpolation but follows a similar performance trajectory. This suggests that the main bottleneck in full tensor interpolation lies in the eigenvalue interpolation step, which—like the component-wise case—relies on RBF methods.

## 4.3 Magnetic gradiometry from Geyer

Finally, we validated the method on real airborne magnetic gradiometry data from Geyer, located in Germany's Erzgebirge region—part of the Central European Variscides. The area features high- and medium-pressure metamorphic units, orthogneiss domes, and post-orogenic granites (Kroner and Romer, 2013), with abundant ore-forming skarns containing magnetic minerals (Burisch et al., 2019; Lefebvre et al., 2019) as well as magnetite-rich quartzites and amphibolites that occur as intercalations within the metamorphic rocks. These rocks contribute to complex magnetic anomalies ideal for real-world evaluation.

We test the ENF method on a real airborne magnetic gradiometry dataset from Geyer (Fig. 6), acquired by Supracon AG in 2016 as part of the E<sup>3</sup> (ErzExploration Erzgebirge) project. As in the synthetic case, we compare ENF to RBF interpolation.

Due to the complex nature of the signal, we run a 50 model ensemble for the Geyer dataset. Each model uses 64 Fourier features, with four length scales of [220,400,1000,100000]. The number of hidden layers is increased to three, with 1024, 512 and 256 neurons respectively. Each model is trained for 600 epochs, with early stopping triggered after 30 epochs of no improvement. A predefined grid with a cell-size of 20 m is used to evaluate the Laplacian loss, with the Poisson sampling radius starting at 500 m, and going to 200 m. Every fourth flight line is used for training, with the others reserved for testing the interpolation. Since ground-truth grids are unavailable, we assess accuracy using residual analysis and R<sup>2</sup> scores computed for points in the withheld lines.

We plot the residual maps for  $H_{xy}$  on test lines (Fig. 7). While absolute  $R^2$  scores are lower than in the synthetic case—owing to added geological complexity and noise—ENF still achieves more than 30% better performance than RBF across most tensor components, with a whopping increase of  $\approx 157\%$  in the  $R^2$  score for  $H_{yy}$ , and an average increase of approximately 57.27%. Both NF and ENF results are better than the RBF across all components. Residuals show that ENF (Fig. 7a) reduces systematic bias between lines and preserves anomaly shapes more faithfully. RBF (Fig. 7b), by contrast, displays patchy behaviour with abrupt shifts between lines—a well-known artifact of interpolating sparse or anisotropically sampled data (Hillier et al., 2014; Wittwer, 2009). The loss curves (Fig. 7d) show a similar trend to the synthetic case, with the data loss terms plateauing around 0.1 nT m<sup>-1</sup>, and the Laplacian loss steadily decreasing.

To get a qualitative overview of the overall result, we plot the gridded visualisations of the  $H_{xx}$ ,  $H_{xy}$ , and  $H_{xz}$  tensor components (Fig. 8). We also compute a result from using all of the flight lines with an RBF interpolator (Fig. 8a-c), serving as our ground-truth. The RBF results from using every fourth line (Fig. 8d-f) reveal strong aliasing and inconsistent behaviour between flight lines—hallmarks of inadequate cross-line interpolation. In contrast, the ENF interpolations (Fig. 8g-i) exhibit smoother transitions and clearer structural trends, especially in directions orthogonal to flight lines. The ENF model successfully mitigates high-frequency striping and captures geologically meaningful features.

## 5 Discussion

#### 5.1 Accurately reconstructing tensor fields

The proposed Neural Field (NF) Interpolator has shown remarkable success in interpolating tensor gradiometry data. Our results show that the additional information contained within the hessian tensor can help derive a more accurate reconstruction of the entire field as sampling gets sparser (Fig. 5), provided the interpolation algorithm can access the full tensor constraints. For equivalent inputs, the NF interpolation recovers a signal that better fits all the tensor components, while maintaining the integrability and physical properties inherent to a hessian tensor field.

We also see equivalent results from all methods when line spacings are tight (i.e., for line spacings of 80 m, 100 m and 120 m in our synthetic tests). This suggests an oversampling with respect to the spatial frequencies in the signal, such that all the interpolation methods converge to the same (correct) result to yield high accuracy metrics. Results then diverge as line spacing increases to 200 m, indicating the neural field interpolation is able to leverage information in the shape of the tensors to continue to derive accurate reconstructions, while the RBF and quaternion methods cannot.

The reason that the results converge with close spatial sampling could be attributed to the equivalence of SLERP and standard linear interpolation as the angle between the quaternions describing the orientations of the input data points goes to zero. Since a tighter line spacing ensures a smoother graduation of the eigenvector orientations (i.e., a smaller change in the angle between the corresponding quaternions), the resulting interpolation is closer to what one would achieve with standard linear interpolation of the components. But, under sparse sampling conditions, the differences seen in the results indicate that an interpolation using neural field formulation better preserves the shape of interpolated tensors, without the need for cumbersome quaternion formalisms.

The interpolated tensor components for Geyer (Fig. 8) also showcase significant improvements over the component-wise interpolation of these tensors. The extension and continuation of the trend from the centre of the grid, towards the north-east is preserved in the ENF result, but is completely absent in the RBF result. Any interpretation of these grids would thus result in significantly different geological structures, highlighting the necessity for appropriate interpolation methods. The Laplacian constraint is handled with an objective minimisation approach in our method. One could potentially enforce harmonicity by design, however this is challenging for 3D (i.e. geophysical potential) fields and difficult to enforce through the nonlinear activation functions inherent to neural networks. In 2D, holomorphic functions (i.e., complex-differentiable functions of multiple variables) consist of real and imaginary parts that are harmonic functions, a fact that is utilised by Harmonic Neural Networks (e.g., PIHNNs; Calafa et al., 2024) to yield exactly harmonic outputs. These concepts do not directly extend to 3D, promoting an objective driven enforcement of the constraint. Vector potential based formulations (e.g. CurlNet; Ghosh et al., 2022) enforce divergence-free fields but fail to enforce the zero curl constraint. Furthermore, as our network consists of non-linear activations, and as non-linear compositions do not generally preserve harmonicity (Chen et al., 2010), we are further motivated to rely on our new mapping that has harmonic elements (see Section 3.2) and use an objective to constrain the Laplacian.

## 5.2 Recovery of vector fields

Many analysis methods applied to tensor gradiometry data require a domain-wide integral to estimate the underlying vector field. The simplest way of computing this integral is by ignoring everything but the last row of the gradiometry tensor, and using the  $H_{xz}$ ,  $H_{yz}$ , and  $H_{zz}$  components to get vector components. Due to the Fourier domain properties, vector components are defined as a vertical integral in the Fourier domain (Mickus and Hinojosa, 2001). Similarly, the power spectrum of these signals can also be used to generate vector components, using transfer functions that fit all of the signals while minimising noise (Vassiliou, 1986). However, in our method, we can completely avoid this potentially complex integral. We can use automatic

differentiation to acquire the vector field components from the predicted scalar potential as the neural field predicts scalar potential and not the gradiometry tensor itself. Importantly, we thus estimate the vector field components exclusively from real measurements, rather than from an integral over a regularly spaced (i.e. interpolated) grid that is already one-step removed from the data.

To test the recovery of vector components from our model, we compared it to the benchmark generated using the RBF interpolation on all flight lines and then applying Fourier domain transfer functions to compute the integral. We also use the transfer functions on the RBF interpolation results for our training data for a baseline comparison (Fig. 9). Comparing the resulting  $b_x$  (Fig. 9, Panels a, d, g) components, we see that features present in both the ground truth and the ENF results are completely erased from the RBF result. Similarly, the shape of the anomaly at the top-right corner of the grid is distorted in the RBF result, but completely preserved within the ENF grid. Slight changes in trend directions (i.e., the shift of the strike of the anomalies to having a smaller azimuth) also cannot be seen in the RBF results, which has prominent "boudinage" artefacts along the flight lines that cause a loss of trend and directional information perpendicular to the flight line. We suggest that these results highlight the ability of the neural field interpolation to extract sensible information (resembling the ground truth) from data acquired at four times the line spacing.

## 5.3 Uncertainty analysis and ensemble models

We also used the stochastic nature of our feature embeddings to do a preliminary uncertainty analysis for the results from our interpolator for the Geyer dataset (Fig. 10). The standard deviation plot shows higher variability in model predictions across regions without data points (i.e., between the flight lines), which could be interpreted as an uncertainty measure. Interestingly, the variance between flight lines seems to scale with the value of the underlying tensor component, leading to heteroscedasticity in the predictions. This might need correction in future developments of our methodology. It is also worth noting that the NF approach has parallels to the turning bands and spectral methods to simulate random fields (Mantoglou and Wilson, 1982), suggesting that a deeper stochastic link to other Gaussian process methods may be possible. This link could be exploited to better understand the variance of neural field ensembles or consider future modifications of the present NF algorithms towards tuned frequency matrix distributions.

The variance of our ensemble model is generally higher for the components with two derivatives in the same dimension (i.e.,  $H_{xx}$ ,  $H_{yy}$ , and  $H_{zz}$ ), and for the derivatives involving the z component (i.e.,  $H_{xy}$  seems to be the least uncertain). High same-dimension double derivative uncertainties might reflect the propagation of uncertainty through differentiation, as uncertainties in two variables have a chance of cancelling out, but are only amplified with multiple passes through the same derivative operator (Li and Oldenburg, 1998). The high uncertainty in the z components likely reflects the lack of information in the z direction, as all of our training data are close to co-planar. Furthermore, we also see that the uncertainty in the recovered vector components (Fig. 10, Panels g, h, i) never goes to zero (even where we have measurements of the tensor), reflecting the lack of information on the constant of integration.

Interpolated grids alter the observation error model: smoothing and continuation introduce spatially correlated errors that, if ignored, can bias ensemble-based inversions (EnKF). Best practice is naturally to invert at the real measurement locations, however when a grid is needed we suggest that our ENF ensemble could provide a mean and a sample covariance for the pseudo-observations. It is possible (although untested) that this might be used as the observation-error covariance in the inversion.

## 5.4 Challenges and future directions

We suggest that the proposed approach opens the door to using neural fields for potential field geophysics, and broader applications involving tensor quantities (e.g., stresses and strains). However, further work and research is needed in several areas. Firstly, our model is highly sensitive to the length scales chosen for the Fourier encoding. As shown by Tancik et al. (2020), optimisation algorithms fail to tweak these scales, meaning they need to be selected with careful empirical tuning. Furthermore, while we have utilised a real-time updating hyperparameter based on the magnitude of the loss, research into other possible avenues of automatising hyper-parameter tuning could boost the usability of our method and help to ensure robust results.

In addition, while the recovery of integrated vector fields is a big advantage of our approach, these have arbitrary integration constants. This ambiguity means that, for every vector component, there is a constant that is unbounded in the other two dimensions. The same problem occurs when we use the Fourier domain transfer functions, as a fundamental lack on long wavelength information leads us to misrepresenting the baseline for the recovered vector field (Ugalde et al., 2024). However, in our methodology, this could be resolved with a few measurements of the vector components included as constraints on the neural field. Therefore, one additional future direction would be to include multiple datasets (e.g., TMI measurements for magnetic gradiometry, satellite or ground gravity measurements for gravity gradiometry) during the training process. Further research on the propagation of uncertainties through our model, as well as impact of ensembles during inversion, would help in improving the robustness of our proposed framework.

Finally, the inclusion of a harmonic decaying term in the feature mapping makes our method a possible contender for an innovative downward continuation scheme, and thus help with the problem of noise amplification in the ill-posed downward continuation of potential field anomalies. This application needs further research, with proper tuning of the weight matrices and data acquired at multiple elevations for validation.

## 6 Conclusion

We introduce an innovative interpolation method tailored to tensor gradiometry data in potential field geophysics. This approach leverages the inherent physical relationships among tensor components by representing them as derivatives of an underlying scalar potential field. Our method clearly demonstrates advantages over conventional interpolation techniques, particularly in scenarios involving sparse and anisotropic data coverage, as are typical during aerial surveys.

Our method has shown substantial improvements in interpolation accuracy, structural fidelity, and robustness against data sparsity during evaluations on both synthetic gravity gradiometry data and a real-world magnetic gradiometry dataset from Geyer, Germany. Quantitative comparisons using metrics such as  $R^2$  scores and Structural Similarity Index Measure (SSIM) highlights the NF interpolator's performance across all tensor components, a preservation of geological trends that are typically used during interpretation, and a reduction of common artefacts caused by line-to-line inconsistencies.

Furthermore, by incorporating stochastic random Fourier features, our model likely opens the possibility to quantify uncertainty. Our analysis reveals heteroscedastic behaviour in the interpolations, and also highlights regions that require further data acquisition or refinement. Additionally, our approach seamlessly integrates vector and scalar field reconstructions through automatic differentiation, simplifying subsequent geophysical analyses and interpretations.

Overall, we argue that the proposed neural field interpolation method represents a significant advancement in processing tensor gradiometry data. Future developments should focus on larger scale applications, better understanding uncertainty of the model predictions, extended vertical interpolation capabilities (e.g., up- and downward continuations), and the integration of this approach into broader geophysical inversion and interpretation frameworks.

Acknowledgements. The authors gratefully acknowledge the Federal Institute for Geosciences and Natural Resources (BGR) and Supracon AG for providing the airborne full-tensor magnetic gradiometry dataset from Geyer, acquired in 2016 as part of the E<sup>3</sup> (ErzExploration Erzgebirge) project. This research was supported by funding from the European Union's HORIZON Europe Research Council and UK Research and Innovation (UKRI) under grant agreement No. 101058483 (VECTOR). The authors thank Ítalo Gonçalves, David Nathan, and one anonymous reviewer for their constructive comments which helped tremendously to improve the manuscript. AVK would also like to thank Vinit Gupta, Ralf Hielscher, and Parth Naik for their valuable insights and delightful discussions, which significantly contributed to improving the clarity and depth of the manuscript.

Code and data availability. Tensorweave is an open-source Python library licensed under the GNU Public License v3.0. It is currently hosted on https://github.com/k4m4th/tensorweave, and the version associated with this publication (including all data and code needed to reproduce our results) is archived at https://doi.org/10.5281/zenodo.16947831 (Kamath, 2025). Notebooks to replicate the figures presented in this paper can be found within the Notebooks directory of this repository. The python environment (packages and specific versions) used is specified in the tensorweave.yml file. The synthetic gravity gradiometry dataset (with all the different line spacings), along with the synthetic density model and the magnetic gradiometry data from Geyer, Germany, can be found in the Datasets directory (in readily readable CSV format). The pre-trained ensemble for Geyer can be found at http://doi.org/10.14278/rodare.3943.

Author contributions. AVK: Conceptualisation, Formal analysis, Investigation, Methodology, Visualisation, Writing – original draft; STT: Conceptualisation, Methodology, Writing – original draft, review & editing; HU: Writing – review & editing; BM:
 Writing – review & editing; RTD: Discussion, Writing – review & editing; MK: Writing – review & editing; RG: Writing – review & editing.

Competing interests. The authors declare that they have no conflict of interest.

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

## **Figures**

**Figure 1.** Synthetic subsurface model and corresponding gravity gradiometry data. (a) Horizontal cross-section of the synthetic geological model at a depth of 140 m, with high-resolution observation points shown as black dots. The five independent components of the gravity gradiometry tensor generated via forward modelling using SimPEG are also shown (b-f). Each panel displays both the high-resolution dataset (grey scale; cell size of 25 m) and the low-resolution dataset (colour; 200 m cross-line spacing and 15 m inline spacing) for the corresponding tensor component.

**Figure 2.** Neural Fourier Field model architecture. The blue block projects the input position vector into a feature space and passes it through the fully-connected layers (orange block) to acquire the scalar potential. The red arrows signify the use of automatic differentiation to acquire the first (gradient) and second (curvature) spatial derivatives of the potential.

Figure 3. Quantitative comparison of interpolation performance for the synthetic dataset. Spatial distribution of residuals between the true and predicted  $H_{xy}$  tensor component using (a) the Truncated Radial Basis Function (RBF) method and (b) the Ensemble Neural Field (ENF) approach (with 25 models in the ensemble). Insets show 1:1 parity kernel density plots comparing predicted and true values. (c)  $R^2$  scores for each tensor component ( $H_{xx}$ ,  $H_{xy}$ ,  $H_{xz}$ ,  $H_{yy}$ ,  $H_{yz}$ ,  $H_{zz}$ ) across three interpolation methods: RBF, the mean of the individual Neural Field (NF) scores from the models within the ensemble, and ENF. The ENF and NF models consistently achieve higher accuracy across all components, while RBF exhibits reduced performance. The loss curves (d) for various components of the loss show the data fitting parts (solid lines) plateau while the Laplacian part (dotted line) keeps decreasing owing to increased sampling with each progressive epoch.

Figure 4. Comparison of gravity gradiometry tensor components derived from two interpolation methods applied to the synthetic dataset. The ground truth  $H_{xx}$ ,  $H_{xy}$ , and  $H_{xz}$  components (a-c) are compared with the results from the Truncated Radial Basis Function (RBF) (d-f) interpolation with 250 nearest neighbours and a smoothing factor of 100, and corresponding results produced by the Ensemble Neural Field (ENF) method (g-i) with 25 models in the ensemble. Black lines in interpolated results (d-i) indicate the input flight lines used for interpolation.

Figure 5. Accuracy metrics as a function of increasing line spacing for the synthetic dataset.  $R^2$  Score (a), Structural Similarity Index Measure (SSIM) (b), and Mean Squared Error (MSE) (c) were computed between the ground truth and the gridded results from the interpolation methods. The Radial Basis Function (RBF) used 250 nearest neighbours, with a smoothing factor of 100, and the Neural Field (NF) model used the same architecture as discussed in Section 3.3. The full tensor interpolation algorithm (QUAT; Fitzgerald et al., 2012) was also included for comparison, using the aforementioned RBF for the eigenvalue interpolation, and SLERP for rotational interpolation. The shaded regions show the minimum and maximum metric across all the components, and the plotted line shows the root-mean-squared metric computed across the components.

**Figure 6.** Case study site near Geyer, Erzgebirge, Germany. Flight lines from a subset of the airborne magnetic gradiometry survey (a), with every fourth line (red) used as input for interpolation and the remaining lines (black) reserved for validation. (b) Spatial distribution of the measured *zz*-component of the magnetic gradiometry tensor across the survey region.

Figure 7. Quantitative comparison of interpolation performance for the Geyer dataset. Spatial distribution of residuals between the true and predicted  $H_{xy}$  tensor component along the test flight lines using the Truncated Radial Basis Function (RBF) (a) method and the Ensemble Neural Field (ENF) (b) approach (with 50 models in the ensemble). Insets show 1:1 parity kernel density plots comparing predicted and true values. (c)  $R^2$  scores for each tensor component ( $H_{xx}$ ,  $H_{xy}$ ,  $H_{xz}$ ,  $H_{yy}$ ,  $H_{yz}$ ,  $H_{zz}$ ) across three interpolation methods: RBF, mean of the individual Neural Field (NF) scores from the models within the ensemble, and ENF. The ENF and NF models consistently achieve higher scores across all components, while RBF exhibits reduced performance. The loss curves (d) for various components of the loss show similar characteristics to the synthetic loss curve.

Figure 8. Comparison of magnetic gradient tensor components interpolated onto a uniform grid (cell size  $= 20 \,\mathrm{m}$ ) using two methods. Gridded  $H_{xx}$ ,  $H_{xy}$ , and  $H_{xz}$  components obtained using the Truncated Radial Basis Function (RBF) interpolation method, with 250 nearest neighbours and a smoothing factor of 100 for all of the flight lines (a-c) are used as the ground truth. We compare the ground truth with the corresponding components interpolated with RBF using every fourth flight line (d-f), and the corresponding components interpolated using the Ensemble Neural Field (ENF) approach (g-i) with 50 models in the ensemble. Each column visualises a distinct component of the tensor. Black lines within the plots indicate the locations of the input flight lines used in the interpolation process.

Figure 9. Comparison of recovered vector magnetic field components from two interpolation methods, evaluated against a high-resolution reference model. We use vector components  $b_x$ ,  $b_y$ , and  $b_z$  computed using Fourier domain transfer functions applied to magnetic tensor components gridded via the Truncated Radial Basis Function (RBF) from all available flight lines (a-c) as our reference. Fourier domain reconstruction of the vector components obtained using the RBF method on tensor data from the training set of flight lines (d-f), and the corresponding results computed from the spatial derivatives of the scalar field predicted by the Ensemble Neural Field (ENF) model (g-i) are shown. The black lines in each panel represent the flight lines used to generate the corresponding component. Each panel shows the histogram-equalised spatial distribution of the respective vector component across the subset of the Geyer survey area, mapped from 0 to 1.

Figure 10. Uncertainty maps for the 50-model ensemble. The standard deviation computed across 50 models for the  $H_{xx}$  (a),  $H_{xy}$  (b),  $H_{xz}$  (c),  $H_{yy}$  (d),  $H_{yz}$  (e), and  $H_{zz}$  (f) tensor components, and the recovered components  $b_x$  (g),  $b_y$  (h), and  $b_z$  (i) vector magnetic field.