# Peer review of "Tensorweave 1.0: Interpolating geophysical tensor fields with spatial neural networks"

_EGUsphere, 2025_

## Author Comment (AC2)

**Author Response RC1 - Ítalo Gonçalves**

**Tensorweave 1.0: Interpolating geophysical tensor fields using spatial neural networks**

Akshay V. Kamath[1], Samuel T. Thiele[1], Hernan Ugalde[2], Bill Morris[3], Raimon Tolosana-Delgado[1], Moritz Kirsch[1], and Richard Gloaguen[1]

[1]Helmholtz-Zentrum Dresden-Rossendorf, Helmholtz Institute Freiberg for Resource Technology, 09599 Freiberg, Germany.

[2]DIP Geosciences, Hamilton, ON Canada

[3]Morris Magnetics Inc., Fonthill, ON Canada

**Correspondence:** Akshay Kamath (a.kamath@hzdr.de)

Dear Ítalo Gonçalves,

We thank you for your time and effort reviewing the submitted manuscript, and are pleased that you appreciated our results. We have incorporated your suggestions into the revised manuscript, as detailed in the following pages. Please note that to facilitate the evaluation of our revision, the line numbers of the reviewers' comments refer to the originally submitted manuscript while line numbers of our responses refer to our revised manuscript.

Kindest regards,
Akshay Kamath (on behalf of the authors)

**Q1) It would be interesting to point to the reference that coined the term "neural field".**

To the best of our knowledge, the paper that coined the term "neural field" for spatial neural networks is Xie et al (2022). The reference has been added to the text at L20.

**Q2) What was the activation function used in the network? How does it impact the results?**

That's an excellent question. In our experiments with various activation functions, we observed that activations that have stable second (and higher order) derivatives tend to perform better. Commonly used activations (such as ReLU) fail to satisfy the $C^2$ differentiability criteria necessary for multiple backpropagations through the same graph. This resulted in abruptly sharp edges in the interpolated field, reducing the model's ability to fit the measured hessians.

Furthermore, some of the activations with $C^2$ differentiability were found to perform worse than others. For example, the Tanh() activation has extremely small second order derivatives which

quickly get saturated (impacting the rate of convergence). In our studies, we used the Swish (SiLU) (Ramachandran et al., 2017) activation by default, but the Mish (Misra, 2019) activation also showed promising results.

The details have been added in the text in Section 3.4, at L193:

"The MLP block in our model uses non-linear activations for all layers except the output layer. As our framework involves computing second derivatives with AD, activation functions like *ReLU* (which do not satisfy the $C^2$ differentiability criterion) resulted in abrupt edges within the resultant interpolation. Notably, even within the activations that satisfy the aforementioned criterion, some functions performed better than the others. For example, the Hyperbolic Tangent activation function has extremely small second order derivatives which tend to get saturated, impeding convergence. These activations are stable, but not ideal for our models. Among the various activation functions tested, *Swish* (Sigmoid Linear Unit, SiLU; Ramachandran et al., 2017) and *Mish* (Misra, 2019) activations provided the best results."

**Q3) The constraints-as-data approach is effective in practice, but perhaps it would be more desirable to encode the Laplace constraint within the model itself. Any comments on how this could be accomplished? Perhaps with a physically derived activation function and/or constraints on the network's weights. The references below contain examples in the context of Gaussian processes.**

We agree with the reviewer that enforcing harmonicity "inside" the model—rather than only via a residual loss—would be desirable. A classical way to guarantee the solution satisfies Laplace's equation is to represent it in a basis that already solves Laplace's equation, and train only the coefficients. Examples include (i) harmonic polynomials/solid harmonics (e.g., $r^l Y^l_m$ on spherical domains) or other Trefftz-type trial spaces, and (ii) Method of Fundamental Solutions (MFS), which places sources outside the domain so that the interior field is harmonic by construction. These approaches are mathematically clean and enforce Laplace exactly, but they require geometry-aware bases (or source placement), and conditioning can deteriorate as the basis grows or the survey geometry becomes complex/draped.

Furthermore, there are also works trying to develop a hard constraint on the harmonicity:

**On holomorphic/complex-analytic parameterizations in 2-D:**

In the plane any harmonic function is (locally) the real part of a holomorphic function. Recent works exploit this to *bake in* Laplace's equation by construction: **Physics-Informed Holomorphic Neural Networks (PIHNNs)** build complex-valued networks whose outputs satisfy the Cauchy–Riemann conditions, so the (real/imaginary) components are harmonic; they demonstrate boundary-only training for 2D Laplace/linear elasticity. **Harmonic Neural Networks** similarly

realize *exact harmonic* outputs on simply-connected 2D domains using holomorphic activations/layers, and propose extensions to multiply-connected domains. These approaches provide clean *hard* enforcement of harmonicity in 2D, but they do not directly generalize to 3D.

PIHNNs can be found here:
Calafà et al., 2024 ( https://doi.org/10.1016/j.cma.2024.117406)

**On "vector-potential + curl" formulations (divergence-free by construction):**

There is also a line of work that introduces an auxiliary vector potential **A** and sets the target field to *curl*(**A**), which guarantees **divergence-free** outputs (widely used in incompressible flow and electromagnetics). Within ML, **Harmonic Neural Networks** include a *CurlNet* variant that models the electric field as *curl*(**A**); outside that paper, several recent studies in computer graphics and scientific ML similarly maintain a vector potential on grids and take its curl to enforce incompressibility. These methods ensure *div*(**F**)=0 by construction (as **F** = *curl*(**A**)) but they do **not** make the field curl-free—hence they don't by themselves yield a gradient field of a scalar potential unless additional constraints/potentials are introduced, constraints which are usually data-driven. This distinction is exactly the non-uniqueness in the **Helmholtz−Hodge decomposition**, where a field can be modified by a harmonic component (both *div*-free and *curl*-free) without changing those constraints.

CurlNet can be found here:
Ghosh et al., 2023 (https://proceedings.mlr.press/v202/ghosh23b/ghosh23b.pdf)

**On "activation functions / weight constraints".**

While linear combinations of harmonic functions are harmonic, **compositions are not**; thus simply choosing a special activation does not, in general, preserve harmonicity through a multilayer network. Put differently: enforcing Laplace's equation is naturally handled by the **function class** (basis/parametrization) or by a **projection operator**, not by standard pointwise nonlinearities. In 2D there is a helpful special case: the real and imaginary parts of a holomorphic function are harmonic, which motivates complex-analytic constructions on planar domains; however, this holomorphic machinery **does not carry over directly** to higher dimensions. This is also why other parametrisations to enforce harmonicity cannot be carried over into the mapping architecture, as non-linearities within the MLP block would potentially undo the harmonicity constraint in the high dimensional feature space.

**Why we used "constraints as data" in this paper:**

Our goal here was a **reproducible** FTG workflow on irregular survey geometries. Hard constraints via harmonic bases (solid harmonics/MFS) require domain tailoring and careful conditioning;

projection layers require a global Poisson solve per step; symbolic null-space constructions analogous to constrained GPs demand problem-specific algebra. Given these engineering costs, we opted for a **data-centric enforcement** with mapping that has harmonic elements (zero-trace/Laplace residuals plus cross-component consistency).

To hint at these methods, we have added a paragraph into our manuscript in Section 5.1, starting at L363:

"The Laplacian constraint is handled with an objective minimisation approach in our method. One could potentially enforce harmonicity by design, however this is challenging for 3D (i.e. geophysical potential) fields and difficult to enforce through the non-linear activation functions inherent to neural networks. In 2D, holomorphic functions (i.e., complex-differentiable functions of multiple variables) consist of real and imaginary parts that are harmonic functions, a fact that is utilised by Harmonic Neural Networks (e.g., PIHNNs; Calafà et al., 2024) to yield exactly harmonic outputs. These concepts do not directly extend to 3D, promoting an objective driven enforcement of the constraint. Vector potential based formulations (e.g. CurlNet; Ghosh et al., 2022) enforce divergence-free fields but fail to enforce the zero curl constraint. Furthermore, as our network consists of non-linear activations, and as non-linear compositions do not generally preserve harmonicity (Chen et al., 2010), we are further motivated to rely on our new mapping that has harmonic elements (see Section 3.2) and use an objective to constrain the Laplacian."

**Q4) Regarding uncertainty estimation, perhaps it would be simpler to implement a Bayesian neural network, which would incorporate uncertainty by resampling the RFF weights at each iteration of training. The MLP weights could remain deterministic if desired.**

We also tested a Bayesian formulation for the Random Fourier Features, in the form suggested by the reviewer. The approach suggested by the reviewer failed due to the fact that the RFF matrices act as projection bases for our coordinates. If a new bank of weights for the RFF matrix is sampled at each iteration, the projective nature of the transformation results in completely different phases for the sinusoids that follow, resulting in the optimiser jumping around the ever changing loss surface. We also tested with various combinations of warmup periods for the training to capitalize on the data first, before the RFF reshuffling begins, but to no avail. Therefore, to utilise the inherent stochasticity present within the models due to the RFF mapping, we went with the ensemble approach coupled with fixed RFF matrices. This is also the reason why the weights for the RFF matrices are left frozen after initialization, and the length scales are made learnable (optionally), to allow more flexibility.

**Q5) In principle the Laplace constraint could be imposed to RBF as well, as the usual radial basis functions are differentiable. This would allow a fairer comparison of the models. Many works model conservative fields with RBF and Gaussian processes, but to my knowledge they only**

**have gradient constraints.**

We agree that a radial basis function could be used to define the potential such that the derivatives are always harmonic. However we do not consider this as an appropriate benchmark as it would be a methodology development in its own right (as we are not aware of current implementations that do this). However, we do enforce tracelessness into the interpolated RBF results by interpolating only five independent components and computing $H_{zz} = -(H_{xx} + H_{yy})$. The main difference that we aim to highlight is the utilisation of multiple tensor components together improving the interpolation, something that the other interpolators do not do. RBF interpolation of potentials would be a possible avenue, but we have not any open source codes that can interpolate with second derivatives, and consider the development of such a tool to be outside the scope of this paper.

**Minor revisions:**

1. **line 153 - missing parenthesis** Rectified.
2. **line 303 - missing parenthesis** Rectified.
3. **Figure 2 - figure shows (sin, cos) features instead of (sin + phase) as described in the text:** The mathematical notation within the text has been modified to be clearer, and now matches Figure 2. Note that Figure 2 has been updated to show that the length of the feature vector is 2*M* (as both sine and cosine are considered).

---

## Author Comment (AC3)

**Author Response RC2**

**Tensorweave 1.0: Interpolating geophysical tensor fields using spatial neural networks**

Akshay V. Kamath[1], Samuel T. Thiele[1], Hernan Ugalde[2], Bill Morris[3], Raimon Tolosana-Delgado[1], Moritz Kirsch[1], and Richard Gloaguen[1]

[1]Helmholtz-Zentrum Dresden-Rossendorf, Helmholtz Institute Freiberg for Resource Technology, 09599 Freiberg, Germany.
[2]DIP Geosciences, Hamilton, ON Canada
[3]Morris Magnetics Inc., Fonthill, ON Canada

**Correspondence:** Akshay Kamath (a.kamath@hzdr.de)

Dear Reviewer,

We thank you for your time and effort reviewing the submitted manuscript, and are pleased that you appreciated our results. We have incorporated your suggestions into the revised manuscript, as detailed in the following pages. Please note that to facilitate the evaluation of our revision, the line numbers of the reviewers' comments refer to the originally submitted manuscript while line numbers of our responses refer to our revised manuscript.

Kindest regards,
Akshay Kamath (on behalf of the authors)

**Q1) First: the comparison is done with fairly simple methods (e.g. RBFs), but it is common to use equivalent sources for processing (e.g. this would be standard with a Falcon AGG survey). At a minimum, equivalent sources should be discussed as an approach, but ideally, including a comparison with equivalent-source-based interpolation would be valuable.**

We agree that equivalent-source (EQL) methods are widely used in operational gravity and gravity-gradiometry processing, and we have expanded the manuscript to include a focused discussion of EQL approaches and their relevance to full-tensor gravity gradiometry (FTG) [Section 2.2, L76]. We have added the following:

"Another widely used approach for interpolating and transforming potential-field (and gradient) data is the equivalent-source/equivalent-layer method: one replaces the true 3D distribution of sources by a 2D layer of hypothetical monopoles or dipoles beneath the observation surface whose field exactly reproduces the measured data on that surface (Dampney, 1969; Blakely,1995).

In practice the infinite layer is discretised into a finite set of sources and the corresponding dense sensitivity matrix is solved—often with regularisation—to obtain source strengths that honour the physical constraints of potential fields and enable stable continuation and derivative transforms (Hansen and Miyazaki, 1984; Blakely, 1995; Oliveira Junior et al., 2023). This formulation is powerful but computationally demanding for large surveys. Consequently, a substantial literature focuses on accelerating the method by exploiting data geometry and matrix structure: scattered equivalent-source gridding (Cordell, 1992); the "equivalent data" concept to reduce system size (Mendonça and Silva, 1994); wavelet compression and adaptive meshing to sparsify sensitivities (Li and Oldenburg, 2010; Davis and Li, 2011); fast iterative schemes in the space/wavenumber domains (Xia and Sprowl, 1991; Siqueira et al., 2017); and scalable algorithms that leverage the block-Toeplitz Toeplitz-block (BTTB) structure of the sensitivity matrix (Piauilino et al., 2025). Recent machine-learning–inspired variants (e.g., gradient-boosted equivalent sources) further cut memory and runtime for continental-scale datasets (Soler and Uieda, 2021). Open-source implementations, notably Harmonica, provide practical tools for gravity and magnetic datasets using these ideas (Fatiando a Terra Project et al., 2024)."

Regarding a direct comparison in our paper: at present we could not identify a maintained, open-source, out-of-the-box implementation for joint FTG equivalent-source interpolation that we could apply reproducibly to our dataset. The widely used open-source Harmonica library provides EQL and a gradient-boosted variant, but its public API does not support joint inversion of multiple data types (e.g., simultaneously fitting the FTG tensor components), which is required for a fair apples-to-apples FTG comparison. Industrial implementations used in Falcon-style processing are proprietary. Recent academic work on fast EQL for AGG and on convolutional/FFT-based EQL for gravity/magnetics (and a 2025 multi-component FTG variant) does not, to our knowledge, provide open code we can reuse directly. In the interest of reproducibility, we therefore limited our quantitative baselines to well-established, openly available interpolators (RBF, minimum curvature) and provided a detailed discussion of EQL (and its limitations) instead.

**Q2) In practice, the full tensor may not be measured in a survey. For example, it is common for gravity gradiometry to only measure two components, e.g. the Falcon system only measures Gne and Guv, and the rest of the tensor is computed from these values. How would you handle this in your method?**

We agree that the Falcon AGG system directly measures two horizontal curvature components—$G_{ne}$ and $G_{uv}$ with $G_{uv} = (G_{nn} - G_{ee})/2$—with a system called the Horizontal Partial Tensor Gradiometer (HPTG) and that the remaining tensor components are reconstructed in processing. However our contribution focuses on generic tensor gradiometry, and many sensors (including the Lockheed Martin FTG System flown by Bell Geospace, and the SQUID Magnetic Gradiometers) do measure all five independent components of the tensor to avoid additional assumptions that

reduce the degrees of freedom within the system. Both our synthetic and real examples use this kind of dataset.

**Q3) As a comment throughout, it would be helpful to have equation numbers to be able to reference.**

Equation numbers have been added to all equations throughout the manuscript.

**Q4) The equations in sections 3.1 & 3.2 are confusing. The notation is mixed: I understand lowercase bold as vectors, should v_i not be a vector? or is it meant to be an entry of v (in which case the first term W\*r is still a vector, so that would need to be indexed? What is the size of r? is it 3N X 1 or N X 3 ?**

We agree with the reviewer in that the equations jumping between vector notation and indexed notation were reducing clarity and hindering a clear understanding of the mathematics underlying our approach. Therefore, we have switched all the notations within the manuscript to a fully vector based notation. Starting in Section 3.1, the RFF Mapping equation has been changed to:

$$\boldsymbol{\nu}_s = [\sin(2\pi\,\mathbf{W}_s\boldsymbol{r}),\ \cos(2\pi\,\mathbf{W}_s\boldsymbol{r})],\ \text{where } \mathbf{W}_s = \ell_s^{-1}\mathbf{W}^{M\times 3}$$

For each individual length scale, with the subsequent line (L128) explaining the terms in the equation. The size of **r** is 3×1, and the matrix **W** has a size of M×3. This acts on the position vector to return a feature vector of size M×1 (2M×1 after it passes through both the element-wise sine and cosine, and gets concatenated along the feature axis, which has now been clarified at L128). Section 3.2 has been remedied in a similar way (see Q6).

**Q5) Also, it would be helpful to clarify that the number of Fourier features (M) is the same as the number of frequencies.**

We agree with the reviewer, and have added the clarification in Section 3.1, L123.

**Q6) These questions follow from the point about equations in sections 3.1 and 3.2. I don't understand how the sizes in the equation at line 127 work. Ws is size M x N, but now r is in R2, does r only have 2 entries, or is it N x 2 ? I suspect you are treating it as N x 2. Do you then add these together or take a dot product to collapse it to a vector? is N the total number of points? or is N just 3 because it is a 3D vector. Clarifying this would help with the rest of the math in this section.**

The equations shown in the manuscript for the mapping correspond to the mapping acting on a single position column vector, of dimensions N×1 where N corresponds to the spatial dimension. In

section 3.1, since the input is 3D and the mapping is applied to all the dimensions, $\mathbf{r}$ is 3×1 and the $\mathbf{W}$ matrix is $M$×3, resulting in a $M$×1 feature vector. In Section 3.2, we split the input position vector into the horizontal position $\mathbf{r}_{xy}$ (which is now a 2×1 column vector) and a scalar vertical coordinate ($z$). The $\mathbf{W}$ matrix now acts only on the horizontal coordinate vector. This has now been clarified in Section 3.2, L151. Furthermore, to avoid any miscommunication about the norm of the $\mathbf{W}$ matrix and the problems with the application of the decay to the features, we have defined a new vector $\mathbf{K}_s$, which is a $M$×1 vector, which is multiplied by the scalar $z$ before being exponentiated to acquire a decay vector of size $M$. The new equation for the harmonic embedding is given at L158

$$\boldsymbol{\nu}_s = \left[ \sin\left(2\pi\, \mathbf{W}_s \boldsymbol{r}_{xy}\right) \odot e^{-\boldsymbol{\kappa}_s z}, \ \cos\left(2\pi\, \mathbf{W}_s \boldsymbol{r}_{xy}\right) \odot e^{-\boldsymbol{\kappa}_s z} \right]$$

Furthermore, we have also added a paragraph on the logic behind our embedding, from L139 - L150. We hope that these improvements help in clarifying the underlying mathematics. We have also upgraded the notations for all other equations, hopefully increasing readability and clarity.

**Q7) Section 3.4 - Network architecture: are you using activation functions between the MLP layers? If so, what are they?**

We have added a paragraph about the activation functions used in Section 3.4, L193:

"The MLP block in our model uses non-linear activations for all layers except the output layer. As our framework involves computing second derivatives with AD, using activation functions like ReLU (which do not satisfy the $C^2$ differentiability criterion) resulted in abrupt edges within the resultant interpolation. Notably, even within the activations that satisfy the aforementioned criterion, some functions performed better than the others. For example, the Hyperbolic Tangent activation function has extremely small second order derivatives which tend to get saturated, impeding convergence. These activations are stable, but not ideal for our models. Among the various activation functions tested, Swish (Sigmoid Linear Unit, SiLU; Ramachandran et al., 2017) and Mish (Misra, 2019) activations provided the best results."

**Q6) It would be helpful to state the full training problem in section 3.4 – e.g. what are you minimizing over, presumably the weights in W, and you are summing this over all available data points.**

We agree with the reviewer and have upgraded Section 3.4 with the necessary details about our training regimen. Furthermore, we have also explained our hyperparameter tuning choice (starting at L227), along with updates to the Loss equations (Eq 10, 11) to explain the logic we use to train our models. We train the weights within the MLP block of our model, and keep the $\mathbf{W}$ matrix

frozen. This is because **W** acts as a projection matrix for our input vector and encodes the input into the higher dimensional feature space, which then acts as the input for the neural network part of our model. Optimising over the weights matrix **W** would amount to finding the perfect projection that minimises the error between the output field and the measured hessians, which is a much more difficult problem to solve. This has been clarified in the manuscript starting at L238, along with the rest of the parameters used for training, as well as details about the learning rate scheduler, and early stopping criterion.

"We train the MLP parameters (weights and biases) with Adam (Kingma and Ba, 2014), while keeping the random Fourier feature (RFF) encoder fixed after initialisation. The initial learning rate is set to $10^{-3}$ (occasionally $10^{-2}$ when the initial loss scale is large). We apply a plateau scheduler that reduces the learning rate by a factor of 0.8 whenever the loss fails to improve for 20 epochs. Optionally, we also optimise a set of learnable length-scale parameters that modulate the Fourier features; the log-values of these scales are stored as parameters and updated jointly with the MLP. However, the learnable nature of these length scales did not help the model convergence greatly, reproducing results explored by Tancik et al. (2020), which suggested that neural fields fail to suitably optimise these length scale parameters."

**Q6) In section 4, it would be useful to show the loss curves and provide some discussion of the training process – e.g. how many iterations? What do you use for an optimization algorithm? What was the stopping criterion used? Was it the number of iterations or a threshold value for the loss? What are the final values for each component of the loss (e.g. the data fit vs. Laplacian loss – it would also be nice to see how these evolve as a function of iteration)**

We agree with the reviewer in that the loss curves provide a visual understanding of the model's training process. Therefore, we have updated Figures 3 and 7 by adding an additional panel showing all the components of our loss as a function of epochs of training. We trained the models for a specified number of epochs (added in the text at L278 for the synthetic, L322 for the data from Geyer) with an early stopping criterion based on the exponential moving average of the total loss. This has been explained in the manuscript starting at L245. The *Adam* optimisation algorithm was used (and has been specified in the text at L238), and the rest of the information about the loss can be seen in the updated Figures as well.

**Other details:**

1. **line 25: define Random Fourier Features before the acronym (RFF)** Rectified, we removed the mention of the acronym and now define it directly in Section 3.1.
2. **The definition of equation 110 is a bit of abuse of notation in defining Ws; it would be cleaner to state Ws = 1/l_s W.** We agree and have updated the equations accordingly.
3. **Equation at line 123, if you want to stick with vectors as boldface, I suggest bolding k-hat.**

**This equation is a bit odd, because rxy is in R2, and it implies that rz = [0, 0, rz] when you add them together, so the sizes don't match. I get what you mean, but you might be better off stating r = [rxy, rz] or similar.** We completely agree and hope that the updates showcased earlier regarding the notation used solve this problem while increasing clarity and readability.

---

## Author Comment (AC4)

**Author Response RC3 - David Nathan**

**Tensorweave 1.0: Interpolating geophysical tensor fields using spatial neural networks**

Akshay V. Kamath[1], Samuel T. Thiele[1], Hernan Ugalde[2], Bill Morris[3], Raimon Tolosana-Delgado[1], Moritz Kirsch[1], and Richard Gloaguen[1]

[1]Helmholtz-Zentrum Dresden-Rossendorf, Helmholtz Institute Freiberg for Resource Technology, 09599 Freiberg, Germany.
[2]DIP Geosciences, Hamilton, ON Canada
[3]Morris Magnetics Inc., Fonthill, ON Canada

**Correspondence:** Akshay Kamath (a.kamath@hzdr.de)

Dear David Nathan,

We thank you for your time and effort reviewing the submitted manuscript, and are pleased that you appreciated our results. We have incorporated your suggestions into the revised manuscript, as detailed in the following pages. Please note that to facilitate the evaluation of our revision, the line numbers of the reviewers' comments refer to the originally submitted manuscript while line numbers of our responses refer to our revised manuscript.

Kindest regards,
Akshay Kamath (on behalf of the authors)

**Q1) Given that interpolated potential field data often serve as input for geophysical inversion, further discussion of the ENF approach's implications in this context would strengthen the manuscript. Specifically, the observation noted in line 293 suggests potential limitations when applying the method in ensemble-based inversion frameworks, such as the ensemble Kalman inversion. These methods rely on statistical assumptions and error covariance structures that could be influenced by interpolation artifacts or over-smoothing. A brief exploration of how ENF interpolation might influence inversion performance and uncertainty propagation would add valuable context for practitioners.**

This is an excellent point raised by the reviewer. We agree with the reviewer in that the uncertainty associated with the interpolation could potentially impact inversion results between the flight lines. In general, all interpolation frameworks involve some sort of smoothing between data points: The inversion of such interpolated datasets is therefore not recommended in general. Instead, inversion results should only be compared at the points that have measurements, to avoid

this interpolation bias.

However, if e.g., numerical optimations require an inversion constrained by gridded data, then we suggest that our ensemble uncertainties may serve as useful weights for each interpolated grid cell. The propagation of uncertainties through inversion is outside the scope of this contribution, but we have added the following text in Section 5.3, L413, to mention this.

"Interpolated grids alter the observation error model: smoothing and continuation introduce spatially correlated errors that, if ignored, can bias ensemble-based inversions (EnKF). Best practice is naturally to invert at the real measurement locations, however when a grid is needed we suggest that our ENF ensemble could provide a mean and a sample covariance for the pseudo-observations. It is possible (although untested) that this might be used as the observation-error covariance in the inversion."

**Minor revisions:**

1. **Line 123: \hat{k} needs to be defined.** We have updated the notation for all equations, eliminating the need for \hat{k}.
2. **Line 388-389: Please update the reference Laloy et al., 2013. I was unable to find any source for it.** We have replaced the missing reference with appropriate papers for ensemble gravity and magnetotelluric inversions (L263).